# Stress Exposure of Evolved Bacteriophages under Laboratory versus Food Processing Conditions Highlights Challenges in Translatability

**DOI:** 10.3390/v15010113

**Published:** 2022-12-30

**Authors:** Mellissa Gomez, Alexandra Szewczyk, Jake Szamosi, Vincent Leung, Carlos Filipe, Zeinab Hosseinidoust

**Affiliations:** 1Department of Chemical Engineering, McMaster University, Hamilton, ON L8S 4L7, Canada; 2Department of Medicine, McMaster University, Hamilton, ON L8P 1H6, Canada; 3School of Biomedical Engineering, McMaster University, Hamilton, ON L8S 4K1, Canada; 4Michael DeGroote Institute for Infectious Disease Research, McMaster University, Hamilton, ON L8S 4K1, Canada

**Keywords:** virus, adaptational evolution, food safety, stability, bacteriophage application, resistance

## Abstract

Bacterial viruses, or bacteriophages, are highly potent, target-specific antimicrobials. Bacteriophages can be safely applied along the food production chain to aid control of foodborne pathogens. However, bacteriophages are often sensitive to the environments encountered in food matrices and under processing conditions, thus limiting their applicability. We sought to address this challenge by exposing commercially available *Listeria monocytogenes* bacteriophage, P100, to three stress conditions: desiccation, elevated temperature, and low pH, to select for stress-resistant bacteriophages. The stressed bacteriophage populations lost up to 5.1 log_10_ in infectivity; however, the surviving subpopulation retained their stress-resistant phenotype through five passages with a maximum of 2.0 log_10_ loss in infectivity when exposed to the same stressor. Sequencing identified key mutation regions but did not reveal a clear mechanism of resistance. The stress-selected bacteriophage populations effectively suppressed *L. monocytogenes* growth at a modest multiplicity of infection of 0.35–0.43, indicating no trade-off in lytic ability in return for improved survivability. The stressed subpopulations were tested for survival on food grade stainless steel, during milk pasteurization, and within acidic beverages. Interestingly, air drying on stainless steel and pasteurization in milk led to significantly less stress and titer loss in bacteriophage compared to similar stress under model lab conditions. This led to a diminished benefit for stress-selection, thus highlighting a major challenge in real-life translatability of bacteriophage adaptational evolution.

## 1. Introduction

Listeriosis is an infection caused by exposure to the foodborne pathogen, *Listeria monocytogenes*. Although *L. monocytogenes* has a relatively low incidence rate, its high mortality rate (20–30%) [1,2] and its implication in numerous worldwide outbreaks [3,4], makes it a pathogen of significant concern. Listeriosis may be caused by ingesting contaminated foods that do not require cooking prior to consumption, such as milk/milk products, fruit and vegetables, and ready-to-eat meats. Foods at high contamination risk are subjected to processing procedures to prevent the transmission of foodborne pathogens, with subsequent refrigeration being strongly recommended to slow bacterial growth. However, *Listeria* has been shown to be resistant to, or capable of adapting to, several food-preservation methods (e.g., osmotic stress from high salt concentrations [5,6,7], desiccation [8,9], low pH [5,6], UV light [5,7], and heating [7]), and is able to grow under refrigerated temperatures [10].

The use of bacteriophages, viruses that infect bacteria, has shown promise as a method of preventing/tackling bacterial contamination in the food production chain [11,12,13,14]. Bacteriophage, or phage for short, treatments are highly specific in targeting their host bacteria [15], thus minimizing their effect on the natural human biome. Phage preparations lack a specific taste or smell [16], and, because of their high specificity, may be designed so as to not interfere with beneficial bacteria in food products. Therefore, phage biocontrol can be used at every stage of the food production chain, from farm to market to consumer, without interfering with food quality or palatability. Whether applied as a prophylactic or as a disinfectant, phages will be exposed to harsh environmental conditions through physicochemical conditions in the food matrix, the food processing procedure, or during storage. Conditions such as low pH (common in fruits and fruit juice) [17], elevated temperatures (even at a relatively low 65 °C) [18], and desiccation [19] have all been demonstrated to reduce the concentration of viable phage (and hence antimicrobial activity) significantly and drastically. We have previously reported an effective method of stabilizing phages against desiccation in solid form [19,20]. This method, however, cannot be generalized to offer protection against other environmental stressors. In this work, we investigate adaptational evolution as a method to select phage populations resistant to a given stressor. The subjection of phage preparations to cycles of a certain stress condition has been effectively used to generate phage populations that are more resistant to UV light [21] and heat treatment [22,23] compared to the ancestral population. Selection cycles conducted on bacterial cultures infected with phage have resulted in the selection of phages with improved lytic ability such that amplification was observed at temperatures that would have been inhibitory to the wild type [24]. However, many of these studies have been conducted under idealized laboratory conditions and their results have not been confirmed under realistic environments.

Phages have proven to be an effective intervention strategy for controlling *L. monocytogenes*. In 2006, Listshield, a phage cocktail designed to control *L. monocytogenes* [25], became the first phage-based commercial food-safety product to receive FDA approval [26]. Since then, multiple phage products have been commercialized for biocontrol in food. Listex is of these commercial products [27,28,29]. Listex, unlike many other commercial products that are formulated as cocktails, consists of P100 phage as its sole strain. In this study, *Listeria* phage P100 was exposed to three common environmental stressors—vacuum desiccation, heating, and low pH—to select for phages with strong resistance to these stressors. We investigated the reversion rates of stress-selected populations, their genomic sequencing, and their biocontrol ability. Finally, the reversion-tested phage populations were tested under three food applications that exposed phage to the investigated stressor and would thus benefit from increased phage survivability. The survival of the reversion-tested phages was compared to that of the P100 stock under these realistic conditions.

## 2. Materials and Methods

### 2.1. Bacteria and Bacteriophage Strains

The commercially available *Listeria* phage, Listex P100, was purchased from Micreos BV (The Hague, Netherlands). The stock Listex P100 titre contained 2 × 10^11^ plaque-forming units (PFU) per mL as reported by the manufacturer and verified prior to experimentation.

*Listeria monocytogenes* serotype 1/2a (ATCC Number: BAA-2659), which was used as the propagation phage host, and *Listeria monocytogenes* serotype 4b, which was supplied by the Canadian Food Inspection Agency, were both used to construct lytic curves. The genome for the latter has been sequenced and deposited to the NCBI under Ascension number JAJOHJ000000000.

### 2.2. Chemicals

The tryptic soy broth (TSB), agar, Tris base, and agarose used in this research were purchased from Fisher Bioreagents (Ottawa, ON, Canada), while gelatin was obtained from Sigma Aldrich (Oakville, ON, Canada). Yeast extract (YE), magnesium sulfate, calcium chloride, hydrochloric acid (HCl), and sodium hydroxide (NaOH) were purchased from VWR (Mississauga, ON, Canada). TSB (30 g/L) or TSB + 0.6% (*w*/*v*) YE was used as the growth media. Agar plates were prepared using 30 g/L and 15 g/L. Calcium-magnesium (CM) buffer with a pH of 7.5 was prepared from 5 mM CaCl2•2H2O, 20 mM MgSO4•7H2O, 6 mM Tris-Cl (pH 7.5), 0.05 g/L gelatin. The buffer was prepared with MilliQ water, and the media were prepared with deionized water. All buffers and media were steam-sterilized prior to use.

### 2.3. Bacteria Culture, Bacteriophage Propagation, and Quantification

All bacterial *L. monocytogenes* strains were grown in TSB or TSB + 0.6% YE. The cultures were prepared by inoculating from frozen stocks and incubating overnight in a shaking incubator at 37 °C and 180 RPM, and the resultant bacterial concentrations were determined using the standard colony count method. Briefly, we plate bacterial suspension on agar plates, incubate overnight at 30 °C, and count the number of colonies that show up on the plate to determine the number of cell-forming units (CFU).

Phage propagation was performed by growing the phage suspension after exposure on a standard plaque overlay [30] assay using *L. monocytogenes* serotype 1/2a as the phage host. Briefly, 100 µL of an overnight bacterial culture was added to a tube containing 3 mL of soft agar (30 g/L TSB, 5 g/L agarose), and mixed via vortexing. Next, 100 µL of the serially diluted phage was added to the tube and the infected culture was vortexed once more before being poured onto a TSB-agar plate. The soft agar plates were incubated overnight at 30 °C. The resulting plaques were isolated by removing the plaque-containing soft agar layer from the overlay plate with an L-shaped cell spreader and relocating it to a 50 mL culture tube, followed by the addition of 1 mL of CM buffer to elute the phage from the soft agar layer. Chloroform (3 μL) was then added to disrupt the bacterial cell membranes and release any remaining encapsulated phage particles. The phage-containing soft agar was left to elute for 15 min at room temperature before the phage–containing liquid portion was separated from the solids via centrifugation at 7000× *g* for 5 min. The phage suspension was quantified by serially diluting the phage as necessary in CM buffer and plating. The phage samples were quantified using the previously described standard plaque overlay method, where number of plaque forming units (PFU) on each plate was determined after overnight incubation at 30 °C. Exposure trials were then repeated with the newly selected and propagated phage population.

### 2.4. Bacteriophage Selection under Stress and Reversion-Testing

Listex P100 phage from a single stock (2 × 10^11^ PFU/mL) was exposed to one of three environmental stresses—desiccation, elevated temperature, and low pH. The selected phages were then propagated non-selectively and quantified before undergoing the subsequent stress cycle, as illustrated in Figure 1. In total, five cycles of selection were completed, with one cycle consisting of exposure, selection, and propagation. Random samples of the selected phages underwent an additional five unstressed cycles of propagation to assess reversion to previous levels of resistance. The surviving phages were propagated by growing the phage suspension after exposure on a standard plaque overlay assay [30] using *L. monocytogenes* serotype 1/2a as the phage host followed by quantification using the standard PFU count method. Experiments were performed in triplicate to confirm that a selected improvement in survivability could be replicated.

#### 2.4.1. Vacuum Desiccation

A 50 μL aliquot of the stock was added to a microtube and placed in a vacuum desiccator, which was then run continuously for 4 h. Following desiccation, the remaining film in the microtube was re-suspended in 500 μL of CM buffer.

#### 2.4.2. Elevated Temperature

A 100 µL aliquot of the stock phage aliquots was placed in a microtube heater heated to 60 °C for 1 h, following which the phage suspension was cooled.

#### 2.4.3. Low pH

1 M HCl and water was added to 100 µL stock phage to a final concentration of 2 mM HCl and final volume of 1 mL, yielding a pH of 2.65 based on molar concentration. The suspension was then incubated at room temperature for 1 h before being neutralized with equimolar quantities of 1 M NaOH.

The phage samples that were subjected to 5 cycles of desiccation stress, high temperature stress, and low pH stress are subsequently referred to S5D, S5T, and S5pH, respectively. After the fifth selection cycle for each stress condition, the sample was plated using a soft agar overlay and re-propagated in the absence of the stressor five times. These samples were then subjected to the same stress conditions given above and titer loss was quantified to determine whether the phages had reverted to their previously susceptible state. These reversion-tested phage samples are henceforth referred to as R5D, R5T, and R5pH, which correspond to the desiccation, high-temperature, and low-pH stress conditions, respectively. All experiments were performed in triplicate and in succession to obtain biological triplicate experiments. Phage titer before and after stress exposure was reported. Magnitude of improvement of phage survivability was calculated by calculating the titer reduction on exposure to a given stress. This titer reduction was then averaged across the replicates.

### 2.5. Bacteria DNA Extraction and Sequencing

The DNA from the *Listeria* serotype 4b sample of unknown strain was extracted using a DNA elution kit (GenElute Bacterial Genomic DNA Kit; Sigma Aldrich; Oakville, ON, Canada) in accordance with the manufacturer’s protocol. Analysis of the eluted DNA sample with a plate reader (Synergy Neo2; Biotek; Winooski, VT, USA) determined that the *L. monocytogenes* serotype 4b sample contained DNA concentration of 30 ng/uL. The complete genome was sequenced using Illumina MiSeq™ System Next Generation Sequencing (NGS), with a genome coverage of 15×. The genome was assembled using the SPAdes genome assembler [31] and then uploaded to the NCBI under Ascension number JAJOHJ000000000.

### 2.6. Bacteriophage DNA Extraction and Sequencing

A modified phenol-chloroform extraction method [32] was used to isolate phage DNA from the fifth selection cycles and the final reversion cycles. The phage samples were first filtered using a 0.22 μm syringe filter and 500 μL of the filtered sample was used for DNA extraction. To remove residual bacterial DNA, 0.5 μL of 0.5 mg/mL DNase, 0.5 μL of 12.5 mg/mL RNase, and 5 μL of 1 M MgSO4 were added to the samples and allowed to incubate at 37 °C for 1 h. The DNase and RNase were deactivated by heating the samples in a water bath for 10 min at 65 °C. Proteinase K (10 μL of 10 mg/mL), CaCl2 (15 μL of 0.1 M), and SDS (120 μL of 10% *w*/*v*) were then added to the phage samples and left overnight on a 55 °C heating block to break down the phage capsid. One volume of Phenol:Chloroform:Isoamyl alcohol (25:24:1) was added to the treated-phage samples, vortexed briefly, and then centrifuged at 22,000× *g* for 10 min. Following centrifugation, 400 μL of the aqueous layer was extracted in 100 μL increments to ensure that the organic layer was left undisturbed. Next, chloroform was added to the phenol-chloroform aqueous extraction at a 2:1 volumetric ratio. The resultant solution was then vortexed and centrifuged as previously described. Once again, 300 μL of the aqueous layer was extracted in 100 μL increments following centrifugation. 3 μL of 3 M ammonium acetate and a 2:1 volumetric ratio of undiluted anhydrous ethanol, which had been stored at −20 °C, were then added to the second extraction and left to sit for 1 h in a freezer set to −20 °C. Following incubation, the solution was centrifuged at 22,000× *g* for 10 min at 4 °C. The supernatant was then discarded, and the pellet was resuspended in 70% ethanol and stored at −4 °C. After brief vortexing, the solution was centrifuged for a final time and the pellet was isolated. The microtube was then left open in a biosafety cabinet for about 1 h to evaporate any residual ethanol. The remaining DNA pellet was then resuspended in 50 μL of buffer.

In total, seven DNA samples were isolated: the ancestral P100 stock, S5D, S5T, S5pH, R5D, R5T, and R5pH. The DNA from biological replicates were mixed for sequencing. The DNA samples were submitted to the McMaster Mobix Lab for sequencing and genomic analysis. Sequencing was performed using Illumina MiSeq™ System Next Generation Sequencing (NGS), with a read length of 2 × 250 bp and a total coverage of 5% to 0.7% per sample. The genome was aligned, and mutations were identified using the *breseq* genomic analysis pipeline [33]. The identified mutations were then compared to a reference P100 genome obtained from NCBI Accession NC_007610.1 [34]. Fixation mutations were excluded from analysis.

The relevant gene products (gp39, gp40, gp102, gp108) were entered into the NCBI BLASTp system [35] to determine analogous proteins. Only matches with Expect (E) values less than 2.00 × 10^−16^ and % identities of greater than 80% were included. Additionally, there were two critical mutations at positions 95,862 and 95,893 that fell within the noncoding region from position 95,836–95,947. This region was cross-referenced in the BLASTn system through Geneious Prime (San Diego, CA, USA) to search for similar regions in other phages.

### 2.7. Quantification of Bacteriophage Infectivity

The ability of the reversion-tested P100 phage samples to infect *L. monocytogenes* was compared to the ancestral P100 stock through evaluation of plaque morphology and kill curves. These experiments were performed on P100 stock and each of the three biological replicates of the R5D, R5H, and R5pH samples, for a total of ten phage samples.

#### 2.7.1. Efficiency of Plaquing and Plaque Morphology

An efficiency of plaquing assay was conducted on the *L. monocytogenes* serotype 1/2a and 4b samples using standard soft-agar overlay [30] to assess whether the reversion-tested phages exhibited reduced strain infectivity as compared to the ancestral P100 stock. The plaque count exhibited on *L. monocytogenes* serotype 4b hosts was normalized for each phage sample by the plaque count exhibited on the *L. monocytogenes* serotype 1/2a host to determine if host infectivity changed as a result of the selection process. All experiments were completed in triplicate. Plates were imaged and plaque morphology was compared using ImageJ (U.S. National Institutes of Health; Bethesda, MD, USA).

#### 2.7.2. Kill Curves

A growth kinetic profile of *L. monocytogenes* infected with the phages was conducted to compare the lytic ability of the reversion-tested phages to the ancestral P100 stock. Lytic curves were obtained for both the *L. monocytogenes* 1/2a and 4b serotypes. The kill curves were constructed using a plate reader (Synergy Neo2; Biotek; Winooski, VT, USA) to measure bacterial concentration, terms of the culture optical density (OD_600_), over time. Subcultures of the *L. monocytogenes* 1/2a and 4b serotypes were first grown to an approximate OD_600_ of 0.4–0.6 m, such that the bacteria were in exponential growth phase. Bacterial subculture was diluted in TSB media and added to a flat bottom 96-well plate (353072; Corning; Corning, NY, USA). Initial OD_600_ was approximately 0.2, corresponding to 3.1 ± 0.3 × 10^8^ CFU/mL and 2.8 ± 0.1 × 10^8^ CFU/mL for the 1/2a and 4b serotypes, respectively. Phages were subsequently added to the wells at a multiplicity of infection (MOI) of 0.37 and 0.41 for the 1/2a and 4b serotypes, respectively, with three wells of bacterial subculture left uninfected as a control. The optical density was recorded over 14 h to provide a relatively short-term comparison of stock P100 performance against the reversion-tested phages. Readings occurred every 10 min where incubation temperature was set to 30 °C with continuous agitation of the well plate. The generation time of the cultures was determined by fitting an exponential trendline to the exponential growth sections of each curve. The fit, as measured by the coefficient of determination *R*^2^, of these exponential trendlines ranged from 0.94–0.99.

### 2.8. Comparison of Application Performance

#### 2.8.1. Bacteriophage Survival on Stainless Steel Surfaces

The P100 stock and desiccation-selected R5D samples were tested under conditions that mimic the sanitation of food-contact surfaces to determine potential differences in survival. P100 and R5D were diluted to 1 × 10^9^ PFU/mL in media (TSB + 0.6% YE) and 100 µL of each sample was separately applied to a 2 cm × 2 cm sheet of food-grade 316 stainless steel (88885K71; McMaster-Carr, Chicago, IL, USA). Three biological replicates were prepared for both the P100 stock and the R5D samples. The phage treated stainless steel sheets were then left to air dry for 24 h. Recovery of the phage from the stainless steel was based on procedures given elsewhere [36,37]. Briefly, the dried, phage-treated stainless-steel sheets were transferred to a sterile conical tube containing 10 mL of CM buffer and a small amount of sterile glass beads. The conical tube was then vortexed for exactly 2 min. The resulting suspension was then serially diluted. Enumeration of surviving P100 was completed using a small scale soft agar overlay method [38], wherein 20 µL of the diluted suspension was plated onto a *L. monocytogenes* bacterial lawn and tilted to run down the length of the plate. Phage enumeration accounted for sampling dilution.

#### 2.8.2. Bacteriophage Survival within Milk after Batch Pasteurization

The Canadian Food Inspection Agency’s general guideline for the batch pasteurization, also known as low temperature long time pasteurization, of milk-based products with <10% fat recommends processing at 63 °C for 30 min to eliminate foodborne pathogens [39]. The ability of the reversion-tested heat-selected R5T samples and ancestral P100 samples to survive batch pasteurization was compared. All samples were first diluted to 1 × 10^9^ PFU/mL in TSB + 0.6% YE. 100 µL of a given phage sample was subsequently added to 900 µL of commercial 2% milk (purchased from a local grocery store) within a 1.5 mL Eppendorf tube to obtain a concentration of 1 × 10^8^ PFU/mL. This phage concentration is recommended in pasteurized milk for the production of queso fresco cheese [40]. The phage-containing milk samples were then placed within a water bath (76308-896; VWR, Mississauga, ON, Canada) set to 63 °C. A water bath was used instead of a dry bath to simulate relatively fast heating time. After 30 min the samples were immediately transferred to an ice bath to mimic the rapid cooling step of pasteurization. After 15 min the phage samples were then serially diluted and plated using the small-scale soft agar overlay method.

#### 2.8.3. Bacteriophage Survival in Apple Juice

Survival of P100 stock and the acid-stressed R5pH samples was tested in commercially available apple juice (purchased from a local grocery store). The pH of the apple juice prior to experiments was measured to be 3.5 (SevenExcellence pH Meter; Mettler Toledo, Port Melbourne, Australia). Phage samples were first diluted to 1 × 10^9^ PFU/mL within TSB + 0.6% YE media to minimize the influence of any buffers or different volumes. 100 µL of the diluted samples were then added to 900 µL of commercial apple juice to obtain an initial concentration of 1 × 10^8^ PFU/mL. Concentration of the phage within the apple juice was then measured after 1 h and after 6 h using the aforementioned soft agar overlay method. Experiments were repeated thrice to obtain biological triplicates.

### 2.9. Statistical Analysis

All experiments were performed in triplicate. Statistical analysis was performed using a two-tailed paired *t*-test through GraphPad Prism 9 (GraphPad Software; San Diego, CA, USA), where viral and bacterial concentration was log-transformed prior to analysis.

## 3. Results and Discussion

### 3.1. Enhanced Stress Resistance and Low Reversion Rates for Selected Bacteriophage

Listex P100 phage was selected for survival when exposed to one of three environmental stresses—desiccation, elevated temperature, and low pH. The selection process consisted of exposing phage from a single stock to a given stress and subsequently propagating the selected phage.

Figure 2 compares the number of infective phage virions in the P100 stock, stress-selected phage samples, and reversion-tested phage samples before and after exposure to the different stress conditions: desiccation (Figure 2A), elevated temperature (Figure 2B), and low pH (Figure 2C). Infective phage concentration was measured in terms of PFU/mL, where the differences between the concentrations before and after exposure indicate the sample’s resistance to the given stress, with greater reduction indicating lower resistance.

The trend in phage survivability over the selection cycles differed between the stress conditions. The phage samples selected for desiccation resistance (Appendix A) showed significant improvement in survivability to desiccation exposure after only one cycle of selection, with subsequent cycles of selection showing little to no further improvement. On the other hand, the phage samples selected for elevated temperature resistance (Appendix A) showed that the concentration of surviving phage after temperature exposure appeared to increase with every successive selection cycle. However, the trend of improved survivability appears to plateau with further selection cycles. Like the samples selected for desiccation resistance, the sample selected for low pH resistance (Appendix A) demonstrated significant improvement in survivability to low pH exposure after only one cycle (0 log_10_ loss) that was maintained over the successive cycles of selection. The first cycle likely resulted in the inactivation of phages with detrimental mutations for survival against a given stress in all cases.

The ancestral P100 sample, which did not undergo selection beforehand, was clearly the most susceptible to the different stress conditions, with reductions in plaque forming units of 4.7, 5.1, and 2.9 log_10_ following exposure to the desiccation, high-temperature, and low-pH stress conditions, respectively. P100 has been previously reported to experience significant titer loss when exposed to a pH of 3 (>5 log_10_) [17], to a temperature of 65 °C (4 log_10_), which is within pasteurization temperature ranges [18], or to desiccation [19]. Five successive stress cycles were performed to isolate and select P100 phages resistant to either desiccation, elevated temperatures, or low pH. The phages selected via this approach exhibited greater survivability in comparison to the ancestral P100 stock under the same stress conditions (Figure 2). The S5D, S5T, and S5pH samples, which previously underwent 5 cycles of selection for resistance against a given stress, experienced titer reductions of only 1.8, 1.8, and 0 log_10_, respectively.

The selected R5D, R5T, and R5pH phage samples, which are the later generations of the selected samples that have been grown in the absence of stress for 5 cycles, experienced titer reductions of 1.2, 2.0, and 0 log_10_, respectively. These reversion-tested samples exhibited similar infectivity to the selected population, suggesting that the reversion-tested populations did not suffer complete reversion in potential beneficial mutations that made them stress-resistant even in the absence of the stress condition. In summary, all stress-selected and reversion-tested samples showed viable phage retention upon exposure to a given stress that was greater compared to the ancestral P100 stock. Direct comparisons to calculate the difference in magnitude of retention between the selection cycles cannot be made as the concentrations prior to exposure differed between samples. However, in all cases the titer endpoint of the ancestral phage was at least an order of magnitude lower than the stress-selected and reversion-tested phage populations. This result occurred despite the ancestral populations starting with a higher initial concentration, suggesting that the stress-selected, and reversion-tested samples have improved resistance to the relevant stress.

The plaque assay method determines concentration of phage in terms of plaque forming units, and thus our data clearly demonstrated that stress-selected phage populations retain a significantly higher concentration of phage that survive and are infective against *L. monocytogenes* following stress exposure. The greatest improvement in survivability was observed among the stress-selected and reversion-tested low-pH populations (Figure 2C), which showed no statistically significant reduction in titer before and after the reversion test (*p* > 0.05). Conversely, a decrease in viable phage titer was still observed in all samples exposed to desiccation (Figure 2A) or elevated temperatures (Figure 2B). Regardless, improvement in titer retention ensures that there is a larger amount of phage available to control a bacterial contamination. Notably, the potentially elevated P100 concentrations in food products due to increased survivability of the resistant populations is not cause for concern, as P100 only infects bacteria and does not transduce bacterial DNA [41]. Furthermore, an oral toxicity study in rats did not reveal any negative effects resulting from the ingestion of P100 [27].

### 3.2. Genomic Sequences of Stress-Selected Samples and Reversion-Tested Samples

The genomes of the ancestral P100 stock, the stress-selected phages, and the reversion-tested phages were sequenced and aligned prior to comparison with a P100 reference genome [34]. The ancestral P100 stock was included in this analysis to account for potential mutations that occurred due to a bottlenecking effect, wherein genetic drift occurs due to the severe reduction in population resulting in decreased genetic diversity. A total of 28 mutations as compared to the refence genome were determined to be fixation mutations and were thus excluded from further analysis. Figure 3 shows the comparison of the P100 stock and both the stress-selected and reversion-tested samples with respect to the location, type, and frequency of the observed mutations. Mutation frequency is defined here as the percentage that a given mutation occurs within the sample data, where samples consist of DNA picked from three biological replicates. The location and type of mutation are shown in the left axis, where “-” refers to a deletion, “+” refers to an insertion, and “r” refers to a repeated segment. The hypothetical gene protein corresponding to each mutation is annotated on the right axis, according to information published for the P100 genome [27]. At this point the functions for many P100 gene proteins are unknown.

Both the stress-selected phage samples and the reversion-tested samples exhibited comparably improved resistance to a given stress. Therefore, mutations occurring with high frequencies in both the stress-selected and the reversion-tested samples, where the mutations also occur in low frequencies in the ancestral P100 stock, may be responsible for the increased resistance to a given stress. Sequencing was completed on samples that were picked from the three biological replicates. The biological replicates display similar levels of resistance to a given stress phenotypically. However, as they evolve independently of each other, the biological replicates may have selected for different mutations over the selection process, leading to different mechanisms of resistance between the replicates.

There are no mutations with similarly high levels of frequency in both the samples that had undergone 5 cycles of desiccation selection and 5 cycles of reversion (i.e., S5D and R5D), as shown in Figure 3A. The most significant mutation affected gp102, specifically the insertion of an AGGAT sequence at location 92,187. This mutation was present at a frequency of 71% in the reversion-tested sample, R5D, but was present in lower frequencies (10%) for the S5D sample. The higher frequency of this mutation in the later R5D generation might be attributable to a bottlenecking effect. Given that both the S5D and R5D samples demonstrated similar levels of resistance desiccation (<2 log_10_ loss), we would expect that mutations responsible for desiccation resistance in the stress-selected phages (S5D) will be preserved in the reversion-tested phages (R5D). Therefore, no mutations highly correlated to desiccation resistance were clearly identified. The lack of genotype change that correlates with phenotype change may instead point to an increase in phenotypic plasticity, that is, the ability of a genotype to exhibit different phenotypic responses under different environmental conditions. Recently, Schaum et al. [42] demonstrated that the diatom *Thalassiosira pseudonanai* had higher plasticity was exposed to thermally fluctuating environment every 3–4 generations than when grown within stable environments. The authors suggest that the evolution of phenotypic plasticity was responsible for the higher thermal tolerance of the exposed samples as compared to samples grown in the stable environment.

The four highest frequency mutations in the S5T sample were point mutations that affected the gp39 and gp40 gene proteins: position 40,757 (65%), position 40,837 (29%), position 40,840 (47%) and position 41,408 (100%) (Figure 3B). Only two of these mutations are present in the later R5T generation: position 40,757 (31%) and 40,837 (21%). The highest occurring mutation in the R5T generation is an insertion of an AGGATAGGAT sequence affecting gp102 (77%), however, it is present at lower frequencies in the heat resistant S5T sample (7%). There are no singular mutations that occur at very high frequencies in both the S5T and R5T samples. It is possible that the different biological replicates of these samples exposed to high temperatures may have developed distinct mutations affecting the same gene proteins responsible for improved survivability.

Unlike the samples that underwent desiccation and high temperature selection, there are mutations that occur with 100% frequency in all the samples that underwent low pH selection (Figure 3C). Both the stress-selected S5pH and the reversion-tested R5pH sample exhibited mutations affecting gp108 at position 95,995, as well as at positions 95,862 and 98,593—which both fall within a non-coding region from position 95,836 to 95,947—with 100% frequency. Critically, none of these mutations appeared at significant frequencies in the stock sample (≤0.5%). Furthermore, the S5pH and R5pH samples were the only samples to demonstrate no statistically significant loss after exposure to their respective stress. This genomic analysis suggests that there is a clear mechanism for stable low pH resistance that each of the biological replicates selected for during the selection process.

Based on the mutation frequency, a few key locations within the genome appear to have been affected by the selection process for the desiccation-selected phages (gp102), the elevated temperature-selected phages (gp39 and gp40), and the low pH-selected phages (gp108 and a non-coding region extending from position 95,836 to position 95,947). Since the functions of these regions have not been determined for the P100 phage, we searched for highly similar regions in other phages. These phages, the relevant regions, and their functions, where previously published, are given in Appendix A. The results of these comparisons suggested that the mutations affecting the heat-stressed samples may impact a receptor-binding protein/tail-fiber protein (gp39) and/or an assembly chaperone (gp40). The mutation affecting the pH-stressed sample may impact an anti-CRISPR endonuclease (gp108), in addition to the noncoding region. The function of the gp102 region that may be affecting desiccation resistance has not been identified at this time.

Gp39, which we assume to be a receptor-binding/tail-fiber protein based on analogous proteins, has previously been predicted to have a melting temperature of 55–65 °C [18]. Therefore, mutations in this protein that lead to an increase in melting temperature would consequently result in a phage population that is resistant to heat stress. Kering et al. [22] also studied the adaptation evolution of phages with improved thermal stability by subjecting various populations to heating at 60 °C for one hour. Their findings showed that the only heat-resistant mutations in phages CX5-1 and P-PSG-11-1 were point mutations that affected tail-tubular proteins. Additionally, mutations affecting a receptor binding protein/tail-fiber protein and an assembly chaperone, the expected function of gp40, were previously found in *Listeria* phages that were successfully adapted through laboratory co-evolution to infect generally phage-resistant rhamnose-deficient *L. monocytogenes*, thus improving host range [43].

The genomic correlations in this study are inconclusive; therefore, further work is required to identify and characterize the mechanisms of resistance. The benefits of mutations are very clear with respect to survival, but the disadvantages are not necessarily evident in the growth phase of propagation. Thus, an efficiency of plaquing assay and lytic curves were used to assess reproduction ability.

### 3.3. Antibacterial Potency Preserved in Reversion-Tested Bacteriophage

The lytic behaviors of the P100 stock and the reversion-tested selected phage samples were assessed via a comparison of plaque morphology, and the construction of a kill curve for two representative strains of *Listeria* serotypes 1/2a and 4b. The serotypes 1/2a and 4b are responsible for the significant majority of the clinical cases of listeriosis [44,45,46].

The *L. monocytogenes* serotype 1/2a strain used in this study was obtained commercially. However, the *L. monocytogenes* serotype 4b strain was gifted. This bacterial strain was sequenced in an attempt to characterize it. Multilocus Sequencing (MLST) [47] determined that the sequence type of *L. monocytogenes* serotype 4b was 2. The assembled genome was also entered into the NCBI BLASTn system [35] to identify any similar strains. The results of this analysis revealed that the *L. monocytogenes* serotype 4b sample genome possessed strong similarities to *L. monocytogenes* strains 02-6679 (NCBI Ascension number NZ_CP008821.1) and 02-6680 (NCBI Ascension number NZ_CP007462.1), strains that have been isolated from stool and cheese, respectively.

Plaque morphology of the *L. monocytogenes* serotype 1/2a and 4b samples infected with the P100 stock and the reversion-tested phage samples is shown in Figure 4. Overall, plaque morphology is consistent between the reversion-tested phage strains and the ancestral P100 stock for both tested *Listeria* strains. All plaques are rounded, completely clear, exhibit well-defined edges, and are approximately the same size for a given *Listeria* strain. The heat-selected R5T sample is an exception as its plaque size appears slightly larger than all other samples. The results of an efficiency of plaquing assay are given in Appendix A, where *L. monocytogenes* serotype 1/2a was used as the reference strain. Infection of *L. monocytogenes* 1/2a was treated as the reference given that the virion count for each sample was unknown. Infectivity of the *L. monocytogenes* serotype 4b sample with the ancestral P100 stock was approximately half that of the infectivity of the *L. monocytogenes* serotype 1/2a sample. Infectivity of the *L. monocytogenes* 4b sample with the reversion-tested phage samples relative to *L. monocytogenes* 1/2a was determined to not be significantly different than that of the P100 stock. The maintenance of plaque morphology in the reversion-tested population suggests no trade-off in lytic ability.

Figure 5A compares the change in optical density (OD_600_) of a *L. monocytogenes* serotype 1/2a bacterial suspension over 14 h when left uninfected and when infected with the different P100 samples OD_600_ is proportional to the concentration of the bacterial suspension. Figure 5B more closely compares the lytic behavior of the P100 ancestral stock and the reversion-tested P100 phages against *Listeria monocytogenes* serotype 1/2a in terms of the change in OD_600_ over time. The uninfected *L. monocytogenes* serotype 1/2a suspension replicates to a very high concentration over time while infection with the P100 stock effectively eliminates the bacteria (Figure 5A). As presented in Figure 5B, the similar lytic trend of the ancestral P100 stock and the reversion-tested phage samples upon infecting the *L. monocytogenes* serotype 1/2a sample demonstrates their strong ability both to suppress the growth of and kill the bacteria in the sample. Furthermore, the maintained low OD_600_ after the elimination of the bacteria indicates that there was no phage-resistant bacterial growth in any of the infected samples over the 14 h monitoring period.

Further analysis of the lytic curves with respect to the maximum OD_600_ (indicative of the maximum bacterial concentration) and the generation time of each infected *L. monocytogenes* serotype 1/2a sample is provided in Figure 5C. All of the lytic curves for the reversion-tested phages showed statistically insignificant maximum OD_600_ values and generation time compared to the ancestral P100 sample.

Growth kinetics of *L. monocytogenes* serotype 4b left uninfected as compared to when infected with the P100 samples is presented in Figure 6A. Similarly, Figure 6B shows the lytic trend difference between the P100 ancestral stock and the reversion-tested P100 phages against *L. monocytogenes* serotype 4b. Comparison with the uninfected bacterial control (Figure 6A) clearly demonstrates that the ancestral P100 stock and all of the reversion-tested P100 phages were able to eliminate *L. monocytogenes* serotype 4b (Figure 6B). Like the lytic curves for the *L. monocytogenes* serotype 1/2a samples, there was no evidence of bacteria-resistant regrowth over the 14 h experiment.

Unlike the *L. monocytogenes* serotype 1/2a experiments, differences in lytic behavior were observed among the different P100 samples that were used to infect *L. monocytogenes* serotype 4b. Although the general lysis trend was similar (limited exponential growth followed by a fast drop in OD), the reversion-tested phages appeared to exhibit better performance than the ancestral P100 stock against *L. monocytogenes* serotype 4b. This improved performance is clearly demonstrated in Figure 6C, which provides the maximum OD_600_ values and generation times for *L. monocytogenes* serotype 4b infected with the different phage samples. The phage samples exposed to desiccation stress (R5D) and low pH stress (R5pH) had a significantly lower maximum OD_600_ than the P100 stock, indicating that these samples are more efficient at eliminating the *L. monocytogenes* serotype 4b bacteria compared to the ancestral phage. No statistically significant differences in maximum OD_600_ were observed between the P100 stock and the heat-stressed samples (R5T). The samples exposed to high-temperature stress (R5T) and low pH stress (R5pH) had significantly higher generation times than the ancestral P100. The average generation rate of the desiccation-stressed samples (R5D) was not statistically significantly different from those of the P100 stock. However, one of the R5D samples had to be excluded from the average generation time calculation as its lytic curve showed bacterial death occurred at a higher rate than bacterial growth almost immediately after infection.

The similarity in lytic trend and maximum OD_600_ between the P100 stock and the reversion-tested phage samples suggest equal biocontrol of the *L. monocytogenes* 1/2a strain. Whereas the infection of the serotype 1/2a sample with the different phage samples exhibited a similar trend in lytic behavior, infections of the serotype 4b sample yielded greater differences between the ancestral phage and the reversion-tested phages. All reversion-tested phages showed either reduced maximum OD_600_ or higher generation times than the ancestral phage. This result suggests that the reversion-tested phages offer improved reproduction and lysis ability against the serotype 4b sample. Overall, these results show that the reversion-tested phages performed similarly or better than the P100 stock in lysing both tested strains of *L. monocytogenes*. Previously, Kashiwagi et al. [48] found that the ancestral phage demonstrated higher thermal stability than reversion-tested phages that were replicated in a bacterial culture held at the inhibitory temperature of the ancestral strain. Nonetheless, our results are consistent with those of other studies [22,23,49] that show no trade-off in lytic ability for phages selected for greater survival. Instead, our results suggest that improved survival under stressed conditions also leads to similar or improved lysis under unstressed conditions.

### 3.4. Comparison of Infectivity for P100 Stock and Reversion-Tested Bacteriophages on Food Preparation Surfaces and in Food Matrices

The survival of the ancestral phage and the reversion-tested phage populations was tested in three food matrices/preparation surfaces representing the selection stresses encountered in this study: desiccation, elevated temperature, and acidic conditions. The results of these application experiments are given in Figure 7.

P100 has been approved for use as a food-contact surface cleaning solution to prevent and treat the proliferation of *L. monocytogenes*. Its application sheet notes that the treatment can work for up to 24 h and recommends that the 2 × 10^11^ PFU/mL phage stock must be diluted to 1% (2 × 10^9^ PFU/mL) prior to spraying onto the relevant surface [50]. The survival of P100 and reversion-tested R5D at an initial concentration of 1 × 10^11^ PFU/mL was evaluated after 24 h of drying on food-grade stainless steel surfaces under ambient conditions. There is evidence in the literature that application of P100 on surfaces within food processing environments can reduce the incidence of *Listeria* [12]. Increased long-term survival of phage on these surfaces will ensure a higher amount of phage is available to act against *Listeria* contamination. We observed that both ancestral and reversion-tested phage populations underwent ~2.5 log_10_ loss (Figure 7A) when air-dried on stainless steel. Leung et al. [19] also reported ~2 log_10_ titer reduction when the same volume (100µL) of P100 was air-dried for 24 h in a well plate under ambient conditions, with complete loss of infectivity after 48 h [19].

Our results in Figure 2 show that the R5D sample had a greater resistance to vacuum drying within a tube for 4 h (~2 log_10_ loss) as compared to the P100 stock sample (~7 log_10_ loss). Vacuum drying of the phages within an enclosed space was chosen as the method of desiccation for adaptational evolution. This method reduced the possibility of contamination during stress cycle exposure as compared to open ambient air drying. However, air drying over a flat surface, the more realistic real-life application scenario tested here, may result in a different level of stress as compared to vacuum desiccation within a tube or air drying within a well plate [19], based on available surface area and drying medium. The mechanistic differences in the drying method may explain why the R5D samples showed improved resistance to vacuum drying compared to the ancestral samples but did not show improved survival to air drying.

Next, we evaluated the survival of P100 stock and the heat-selected R5T sample in commercial milk under conditions that mimic batch pasteurization (exposure to 63 °C for 30 min). *L. monocytogenes* has been implicated in outbreaks linked to products made with pasteurized milk [51]. Post-processing contamination may be limited by the addition of P100 to milk. However, government agencies restrict the addition of ingredients to milk post-pasteurization [52]. Our results show that the P100 stock underwent ~4 log_10_ loss whereas the R5T sample underwent ~3 log_10_ loss when pasteurized (Figure 7B). The difference in survivability between these two samples was determined to be statistically insignificant. Similar results were achieved by Ahmadi et al. [18], which reported that the commercial preparation of P100 underwent ~4 log_10_ loss when held at 65 °C in a circulating water bath for 20 min. We hypothesize that the choice of medium may have affected the selection process. In our phage adaptational evolution experiments, the P100 ancestral stock was selected under successive heat exposure cycles using TSB as a medium. This may have resulted in phage populations optimized for heat shock survival in TSB only. Previously, Komora et al. [53] demonstrated that different food matrices can have protective or harmful effects on the survival of P100 when exposed to high pressure. Therefore, the lack of specificity toward a given food matrices in phage adaptational evolution experimental design may have impaired the R5T sample’s ability to survive batch pasteurization (63 °C for 30 min) in milk.

Finally, we evaluated the survivability of reversion-tested phage in an acidic beverage. Many fruits, such as lemons, blueberries, and apples, are acidic, with citric fruits having the lowest pH. A study conducted by the Canadian Food Inspection Agency found that apples and melons have demonstrated higher incidence of *L. monocytogenes* contamination than other fruits in Canadian retail markets [54]. However, phage P100 has been found to be effective at reducing contamination in melons, but not apples [55]. Therefore, apple juice was chosen as a medium to test the applicability of the phage populations selected for increased survival under low pH conditions. Apple juice was chosen instead of apple slices to prevent influence of desiccation on a solid food matrix on phage survival. Both the P100 stock and R5pH samples underwent statistically insignificant levels of loss after 1 h (Appendix A) and approximately 2 log_10_ loss after 6 h within commercial apple juice (Figure 7C). There was no significant difference in survival between the two samples for any given timepoint. This degree of loss was much less than ~3 log_10_ loss experienced by the P100 ancestral stock after exposure to a pH of 2.65 (Figure 2). The improved survivability of P100 stock after 1 h within apple juice (pH 3.5) may be due to the reduced acidity as compared to the selection environment (pH 2.65). Previously, Fister et al.’s [17] found that the titer of P100 was stable at pH 4 for 24 h but experienced greater than 5 log_10_ loss after only one hour exposure to pH 3 or lower. Their result suggests P100′s lethal acidity is somewhere between pH 3 and 4.

The P100 phage has previously been theorized to have limited applicability in low pH fruits and fruit juices [53,55]. Oliveira et al. [55] tested the ability of P100 to eliminate *L. monocytogenes* in melon (5.77 pH), pear (4.61 pH) and apple juice (3.7 pH). Their results found that P100 was stable in both melon and pear juice when held at 10 °C for 8 days whereas P100 in apple juice underwent 7 log_10_ loss under the same conditions. Our work showed that both the P100 stock and the R5pH sample had similar titer loss patterns within a commercial apple juice at approximately 3.5 pH. Under experimental adaptational evolution (Figure 2), the R5pH sample had shown greater survivability within a 2.65 pH environment for 1 h than the P100 stock. Given these results, it was expected that the R5pH sample would also show improved survivability as compared to P100 when held within acidic apple juice. However, the acidity of the apple juice may not be the only factor affecting phage titer loss. Leverentz et al. [56] included magnesium chloride in the application of a phage cocktail to reduce contamination of *L. monocytogenes* on apples. The magnesium chloride was theorized to improve phage recovery due to neutralization of the acidic environment. However, they found that the addition of magnesium chloride had no effect on phage effectiveness. The results shown in our study further imply that P100 inactivation in apples may be due to factors other than pH, such as the presence of tannins, which have also been linked to antiviral effects [57]. Further work is needed to determine the factors affecting phage loss on apples in order to select against them. Our study indicates that while selecting for phages resistant to a given stress under laboratory conditions is promising, their improved survival does not necessarily translate to realistic food applications. This further highlights the importance of mechanistic understanding of phage infectivity under stress conditions.

## 4. Conclusions

*L. monocytogenes* is an especially pernicious food-borne pathogen due to its ability to survive and grow despite the application of common bacterial-control methods, such as refrigeration, desiccation, heating, and exposure to acidic environments. Commercial phage P100 is able to successfully infect and lyse *L. monocytogenes*. However, P100′s sensitivity to desiccation, high heat, and low pH results in significant loss of infectivity when exposed to these stressors. Using directed evolution, batches of P100 phage was selected for resistance to key stressors common in food processing and/or food matrices, namely, desiccation, elevated temperatures, and acidic environments. Our stress-selected phages demonstrated improved survival up to several orders of magnitude greater than the ancestral population under stress and retained their stress-resistant phenotype even after several passages in the absence of the stressors. Full characterization of the possible mutations responsible for these selected phages’ increased resistance to each stress was hindered by the lack of information on the proteome of phage P100, a limitation that is a major bottleneck in the field of phage biology.

Despite promising results under a laboratory setting, the same benefit over ancestral phage was not translated when the reversion-tested phages were used under more realistic conditions. Specifically, the selected phages did not demonstrate significantly increased survivability over the ancestral phages when dried on food-grade stainless steel, nor when batch pasteurized within milk. This may be due to all the complexities of the stresses encountered within a realistic environment, such as the physical and chemical components of the matrix, that are not replicated during the selection process. The acid-resistant phages also exhibited the same survivability as ancestral phage within apple juice, which may be due to the presence of tannins within some fruits. Even though evolutionary adaptation has garnered more attention in recent years as the means to increase efficacy of phage, our work highlights the challenges affecting the applicability of this method in real life, complex, heterogeneous environments such as food matrices or even within the human body. Therefore, we conclude that investigation into evolutionary adaptation to improve phage efficacy requires more mechanistic investigation.

## Figures and Tables

**Figure 1 viruses-15-00113-f001:**
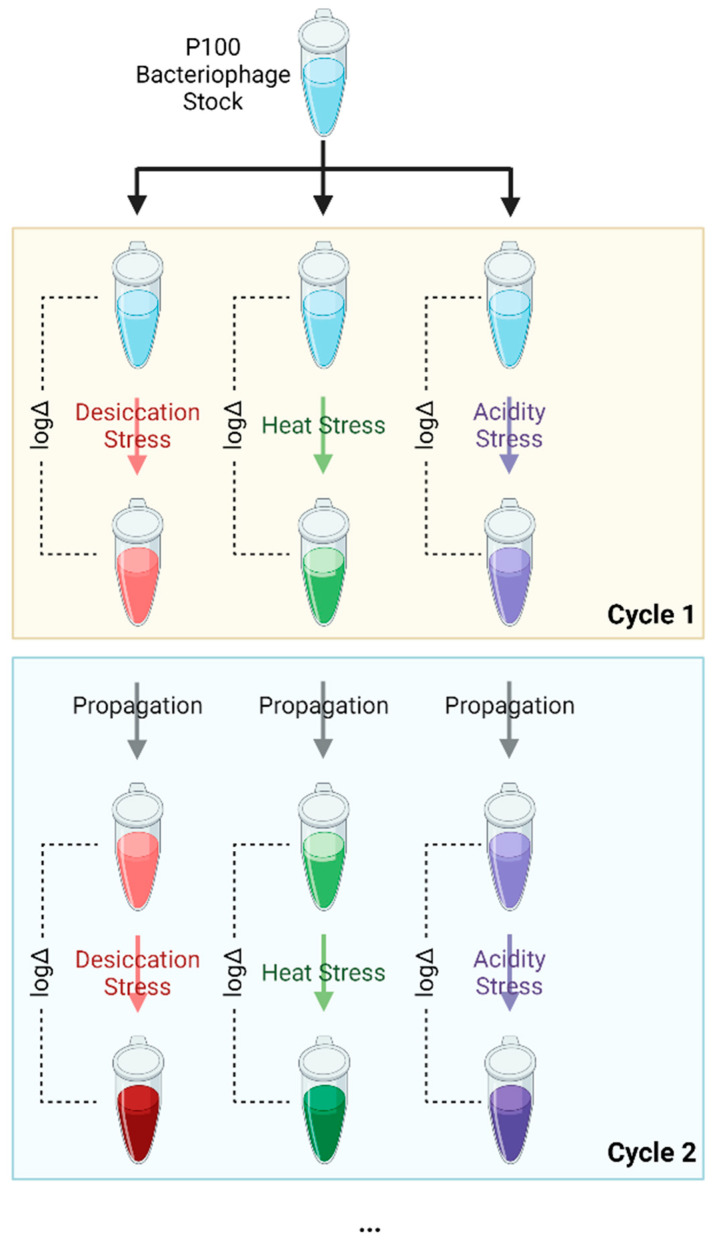
Schematic illustrating the stress-propagation cycle conducted on the P100 bacteriophage samples. The samples underwent five selection cycles to select for bacteriophages that were resistant to either desiccation, heat, or acidic (low pH) conditions. Each color (red, green, and purple) represents a certain collection of random mutations in phage that provide resistance to a certain stressor. Color intensity represents the process of theoretical enrichment of stress resistant phage from the stock of wild type phage which contains a mixture of random mutations. All experiments were completed in three biological replicates. Figure created with Biorender.com.

**Figure 2 viruses-15-00113-f002:**
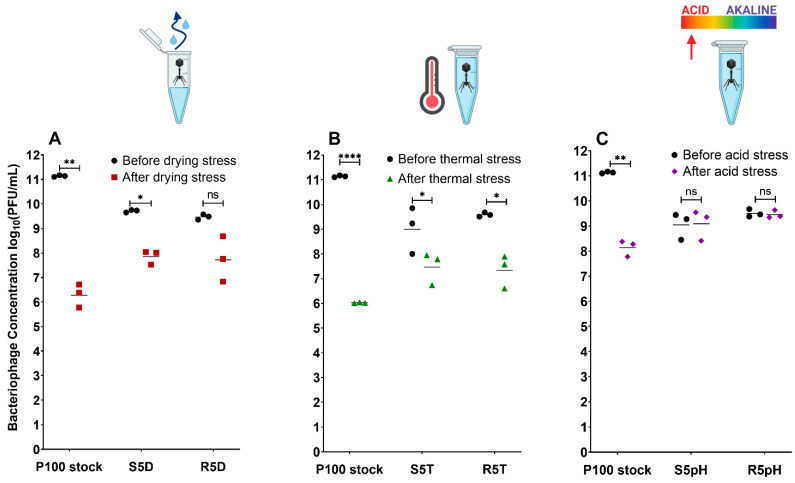
Measured concentrations of viable phage in the P100 stock, a sample that has undergone 5 cycles of selection for a given stress, and a sample that has undergone 5 cycles of selection followed by 5 cycles of propagation without exposure to stress before and after exposure to: (**A**) desiccation for 4 h; (**B**) temperatures of 60 °C of 1 h; and (**C**) estimated pH of 2.65 for 1 h. Results are shown as the individual results of triplicate experiment, where the geometric mean has been marked. Phage concentration prior to exposure presented in black and concentration after exposure presented in colour. Significant differences in phage concentration before and after exposure to a given stress are marked as * *p* ≤ 0.05, ** *p* ≤ 0.01, and **** *p* ≤ 0.0001. Nomenclature: n.s.: not significant; S5D: P100 sample that underwent 5 cycles of desiccation selection; S5T: P100 sample that underwent 5 cycles of elevated temperature selection; S5pH: P100 sample that underwent 5 cycles of low-pH selection; R5D: S5D sample that was propagated 5 times in the absence of stress; R5T: S5T sample that was propagated 5 times in the absence of stress; R5pH: S5pH sample that was propagated 5 times in the absence of stress. Icons from Biorender.com.

**Figure 3 viruses-15-00113-f003:**
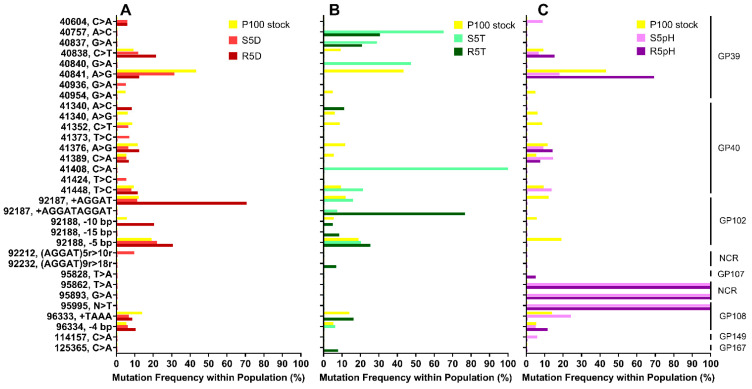
Mutation frequency for the P100 stock (yellow) and (**A**) the desiccation-selected S5D (light red), R5D (dark red) samples, (**B**) the heat-selected S5T (light green), R5T (dark green) samples, and (**C**) the acid-selected S5pH (light purple), and R5pH (dark purple) samples as compared to a reference P100 genome (NCBI Accession NC_007610.1) [34]. The location and type of mutation are shown in the left axis, where “-” refers to a deletion, “+” refers to an insertion, and “r” refers to a repeated segment. The hypothetical gene protein corresponding to each mutation is annotated on the far right. Abbreviations: GP: gene product; NCR: non-coding region. Nomenclature: S5D: P100 sample that underwent 5 cycles of desiccation selection; R5D: S5D sample that was propagated 5 times in the absence of stress, S5T: P100 sample that underwent 5 cycles of high-temperature selection; R5T: S5T sample that was propagated 5 times in the absence of stress, S5pH: P100 sample that underwent 5 cycles of 5 of low-pH selection; R5pH: S5pH sample that was propagated 5 times in the absence of stress.

**Figure 4 viruses-15-00113-f004:**
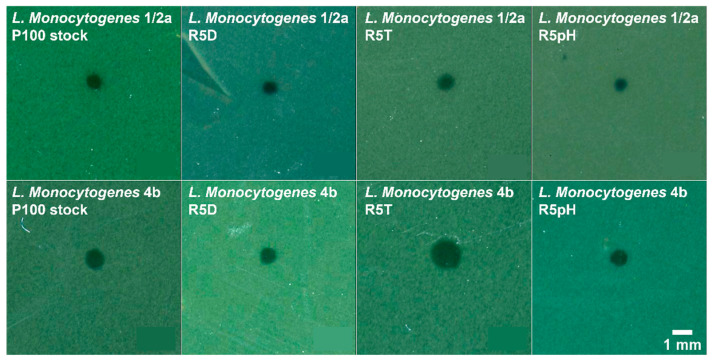
Plaque morphology of the different P100 bacteriophage samples infecting the *L. monocytogenes* serotype 1/2a (**top row**) and 4b (**bottom row**) bacteria samples, where the plaques represent a clearing in the bacterial lawn due to bacteriophage infection. Plaques shown here are representative for each imaged soft agar overlay plate. All plaques are completely clear, round, and display defined edges. Plaque size appears to be similar for infection of a given *L. monocytogenes* strain. Nomenclature: R5D: P100 sample that underwent 5 cycles of desiccation selection followed by 5 propagation cycles in the absence of stress; R5T: P100 sample that underwent 5 cycles of high temperature selection followed by 5 propagation cycles in the absence of stress, R5pH: P100 sample that underwent 5 cycles of low pH selection followed by 5 propagation cycles in the absence of stress.

**Figure 5 viruses-15-00113-f005:**
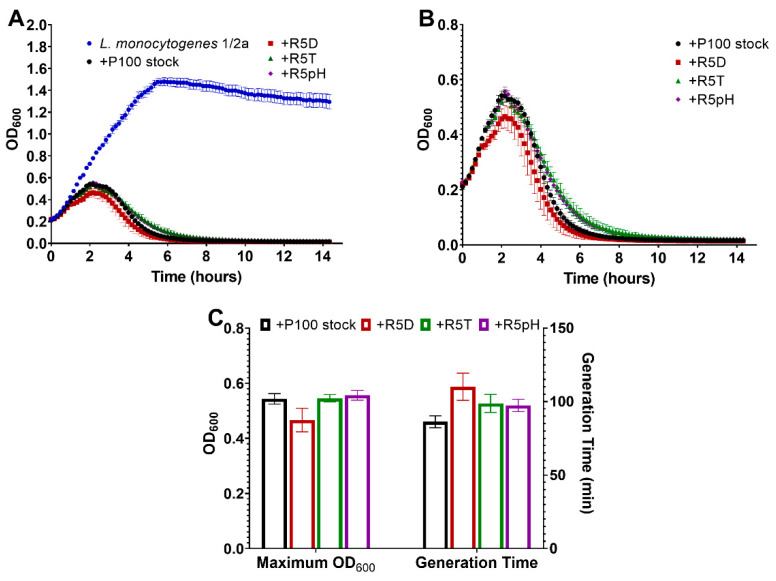
(**A**) Growth curve of a *L. monocytogenes* serotype 1/2a sample left uninfected as compared to infection with P100 stock and reversion-tested P100 samples selected for resistance to desiccation (R5D), resistance to high-temperature exposure (R5T), and resistance to low-pH exposure (R5pH). (**B**) Same growth curve of an *L. monocytogenes* serotype 1/2a sample infected with P100 stock and reversion-tested P100 samples R5D, R5T, R5pH shown at a higher OD_600_ resolution to compare the different growth curve trends. (**C**) Maximum OD_600_ reading and generation time (minutes) for *L. monocytogenes* serotype 1/2a samples infected with different P100 bacteriophage samples. Three biological replicates were tested for the different engineered P100 samples, with all experiments being performed in triplicate. Results are shown as average of the biological replicates ± standard deviation for the reversion-tested phage and the average of the technical replicates ± standard deviation for the P100 stock. Differences in maximum OD_600_ and generation time between the P100 ancestral stock and the reversion-tested samples was determined to be statistically insignificant. Nomenclature: R5D: P100 that sample underwent 5 cycles of desiccation selection followed by 5 propagation cycles in the absence of stress; R5T: P100 sample that underwent 5 cycles of high-temperature selection followed by 5 propagation cycles in the absence of stress; R5pH: P100 sample that underwent 5 cycles of low-pH selection followed by 5 propagation cycles in the absence of stress.

**Figure 6 viruses-15-00113-f006:**
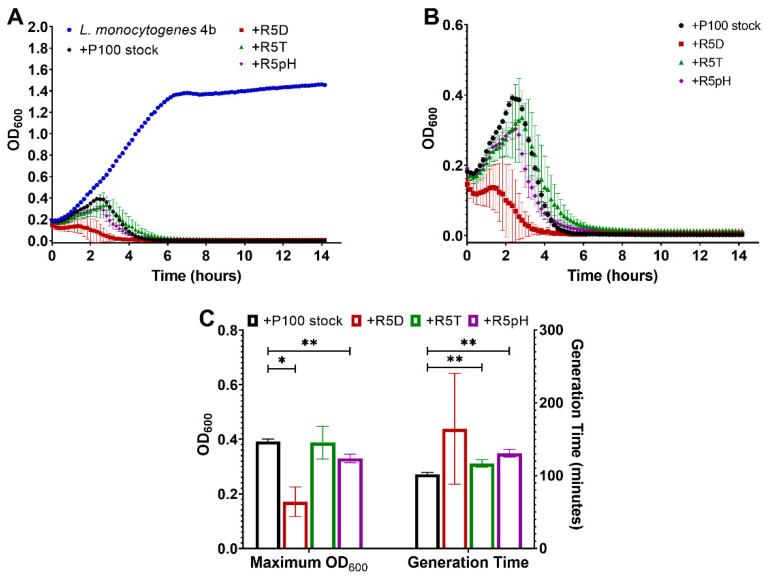
(**A**) Growth curve of an *L. monocytogenes* serotype 4b sample left uninfected as compared to infection with P100 stock and reversion-tested P100 samples selected for resistance to desiccation (R5D), resistance to high-temperature exposure (R5T), and resistance to low-pH exposure (R5pH). (**B**) Same growth curve of an *L. monocytogenes* serotype 1/2a sample infected with P100 stock and reversion-tested P100 samples R5D, R5T, R5pH shown at a higher OD_600_ resolution to compare the different growth curve trends. (**C**) Maximum OD_600_ readings and generation times in minutes for *L. monocytogenes* serotype 4b samples infected with different P100 bacteriophage samples. Three biological replicates were tested for the different engineered P100 samples, with all experiments being performed in triplicate. Results are shown as average of the biological replicates ± standard deviation for the reversion-tested phage, except for the R5D generation time result, and the average of the technical replicates ± standard deviation for the P100 stock. The generation time for one of the R5D biological replicates could not be calculated as there was no measured bacteria growth after infection, thus the results for this sample are given as the average ± standard deviation of two biological replicates. Statistically significant differences compared to the P100 stock are marked as follows: * *p* ≤ 0.05, and ** *p* ≤ 0.01. Nomenclature: R5D: P100 sample that underwent 5 cycles of desiccation selection followed by 5 propagation cycles in the absence of stress; R5T: P100 sample that underwent 5 cycles of high-temperature selection followed by 5 propagation cycles in the absence of stress; R5pH: P100 sample that underwent 5 cycles of low-pH selection followed by 5 propagation cycles in the absence of stress.

**Figure 7 viruses-15-00113-f007:**
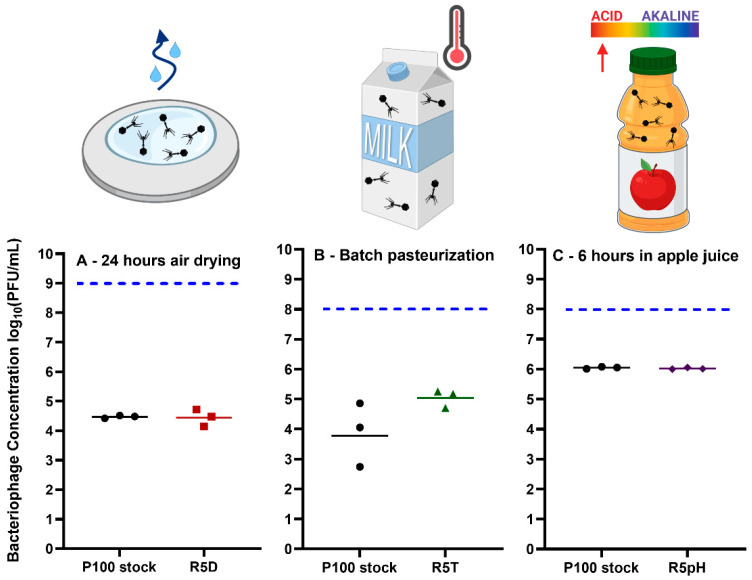
Remaining bacteriophage concentration (PFU/mL), of P100 stock and the reversion-tested bacteriophage samples R5D, R5T or R5pH after exposure to (**A**) 24 h of air drying on a food-grade stainless-steel sheet, (**B**) 63 °C for 30 min within milk (batch pasteurization), and (**C**) apple juice for 6 h. Triplicate results are shown, where the geometric mean is marked as a horizontal line. Black circles represent the P100 ancestral stock, red squares represent the R5D sample, green triangles represent the R5T sample, and purple diamonds represent the R5pH sample. The horizontal dotted blue line represents the initial concentration of the P100 and reversion-tested samples. This starting concentration was selected based on available application sheets for P100, that is, a concentration of 1 × 10^9^ PFU/mL on food preparation surfaces and 1 × 10^8^ PFU/mL within liquid food matrices. Nomenclature: R5D: S5D sample that was propagated 5 times in the absence of stress; R5T: S5T sample that was propagated 5 times in the absence of stress; R5pH: S5pH sample that was propagated 5 times in the absence of stress. Icons from Biorender.com.

## Data Availability

The data presented in this study are available from a repository: 10.5281/zenodo.7246639.

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
