# Peer review of "Stress Exposure of Evolved Bacteriophages under Laboratory versus Food Processing Conditions Highlights Challenges in Translatability"

_viruses, 2022, doi:10.3390/v15010113_

Round 1

Reviewer 1 Report

The manuscript “Discrepancy between stress response under lab conditions and food matrices highlights challenges in translatability of adaptational bacteriophage evolution”  investigates the survivability of stress-adapted bacteriophages in both realistic and laboratory settings. The work within highlights the difficulties in transferring the biological responses observed under standardised laboratory conditions to that of “real world” environmental conditions, which will be of interest to a wide audience. In addition, the authors show that limitations surrounding phage genome annotation and protein function are a blockade to identifying key modes of adaptive resistance. Specific comments and questions are provided below.

L45-53: Please provide appropriate references

L55: Typo? should be as high as 65 oC, not low as stated?

L100: Please define YE

L107&L117: Can detail of the volume(s) of culture/phage used for the propagation be provided?

L111: Please define CM

L121: Please indicate if quantification refers to PFU/ ml etc

L127: Please indicate the titre of the stock used

L131: How were phage for revision selected? Was this random or were set criteria applied?

L143: Unknown if mutations have accumulated at this point in the experimental process?

L155: Please indicate the volume of phage stock used

L156-157: Were the pH values confirmed via accurate reading?

L229: >80% ID – reason for high cut off? Was % cover considered?

L310: pH differs from that of Lab environment testing, was this considered?

L395: P100 stock also shows greater survivability in low pH compared to that of desiccation or thermal stress? Is this a known trait for P100?

L447-496: It appears that the genomic positions listed do not correlate to the gene products as listed in Fig3. For example, according to Fig 3C, the mutations of 100% frequency (95862, 98593, 95995) are indicated as falling within gp39 and not the reported NC region and gp108? Similarly, for Fig3A/B. Please check the mutations and associated ORFs to ensure clarity of explanation for the reader.

L484-497: Did the authors attempt any further analysis on the identified gp(s) of interest such as Hhpred / Pfam etc?

Fig5 and Fig6: Excess of data points on graphs (A&B) making them difficult to view. Could these be reduced for better resolution for the reader?

L619: Can the authors speculate as to the reason for this discrepancy?

L682: Please define LTLT

L705: Typo? Should be R5pH to coincide with Fig S4?

L746 & L754: Typo-stress selected? Results section indicates that reversion phage were used in real condition assays?

Reviewer 2 Report

Comments to the Author

 This manuscript reports on the evaluation of bacteriophage P100 against three stress conditions (desiccation, elevated temperature, and low pH) to select for stress-resistant bacteriophages and evaluate their applicability in food environments.

Title: the title is confusing, please rewrite.

Delete Figure 1 and describe the method in the text.

Section 3.1. “Enhanced stress resistance and low reversion rates for selected phage”: missing discussion.

Section 3.2. “Genomic sequences of stress-selected samples and reversion-tested samples” and section 3.3. “Antibacterial potency preserved in reversion-tested phage”: both sections were not exhaustively discussed.

The manuscript does not highlight the novelty of the work.

English usage throughout the text must be revised (odd expressions).

Reviewer 3 Report

In the article “Discrepancy between stress response under lab conditions and food matrices highlights challenges in translatability of adaptational bacteriophage evolution” a study about the adaptation of phage P100 to different stress factors is performed. It is shown that even though the adapted phages have better performance in laboratory stress settings this is not observed for real conditions. The study is well performed nevertheless there are some changes that should be performed to improve the manuscript.

Some minor changes

Q1. Throughout the manuscript there are references to bacteriophages or phages. To maintain consistency only one form should be selected and used. Please check the manuscript for this.

Q2. Throughout the manuscript there are º instead of °. Please revise the manuscript to change this.

Q3. Ln 100. The YE appears without being stated what it means beforehand. Please add this information.

Q4. Ln 112. Change “were” for “was”.

Q5. Remove pH from Ln 312.

Q6. In the Figure 5 legend change “curve of an” to “curve of a”.

Q7. Ln 609. Add in to sentence before “Figure 6C”.

Major concerns:

Q1 Ln 72 to 74. This sentence is confused, should be reformulated for better understanding.

Q2. Ln 102. Add the rotation used for incubation.

Q3. Ln 103. In summary doesn’t seems the better word for this, considered change for briefly.

Q4. Ln 111. CM buffer is supposed to be SM buffer? From Saline-Magnesium buffer. And the composition only appears latter in the manuscript. Should be described in the first time it is mentioned.

Q5. Ln 156. What temperature was used for this incubation?

Q6. Section 2.5. Why there is methodology for Bacteria DNA extraction in this manuscript? There is no results or discussion for this.

Q7. Throughout the material and methods section there is some discussion. Considered removing the discussion bits. Example ln 225 to 231 or 274 to 280.

Q8. In the plots as it is represented PFU/mL values the Y-axis should be logarithmic. It is also possible to use the log-transformed values used for the statistical analysis as stated in the section 2.9 and maintain the Y-axis in scientific (exponential) notation.

Q9. The comparisons between P100 stock and the different adapted phages should be performed using the same initial concentration. In fact, in the discussion this is mentioned (ln 385 to 387). This should be reformulated. Considered re-do the experiments. In fact, this can be important to guarantee that the adapted phages perform better that P100 stock in laboratory settings.

Round 2

Reviewer 2 Report

This manuscript reports on the evaluation of bacteriophage P100 against three stress conditions (desiccation, elevated temperature, and low pH) to select for stress-resistant bacteriophages and evaluate their applicability in food environments. The manuscript was significantly improved. However, most of the changes indicated with numbers lines in the “responses to reviewers” are very difficult to follow in the manuscript.

Author Response

We are pleased that the reviewer considers the changes made as a substantial improvement. Based on the second response, we have submitted a marked version of manuscript with the relevant edited sections more easily identifiable by blue colored font.

Reviewer 3 Report

The authors have now provided a much improved manuscript. Clearly, this new version will be acceptable for publication after a few minor issues have been solved:

Q1. Section 2.5: There is still some discussion in this section. The authors explain why performing the bacteria DNA extraction, nevertheless this is still not clear in the manuscript. The line 208 to 212 do not belong the this section, they should be included in the results and discussion. 

Q2. Figure 1 should be change so the Y-axis starts at 0 (zero). 
